# Comparative Grain Yield, Straw Yield, Chemical Composition, Carbohydrate and Protein Fractions, In Vitro Digestibility and Rumen Degradability of Four Common Vetch Varieties Grown on the Qinghai-Tibetan Plateau

**DOI:** 10.3390/ani9080505

**Published:** 2019-07-31

**Authors:** Yafeng Huang, Fangfang Zhou, Zhibiao Nan

**Affiliations:** State Key Laboratory of Grassland Agro-ecosystems, Key Laboratory of Grassland Livestock Industry Innovation of Ministry of Agriculture and Rural Affairs, College of Pastoral Agriculture Science and Technology, Lanzhou University, Lanzhou 730020, China

**Keywords:** common vetch, straw, nutritive value, varietal effect, ruminants

## Abstract

**Simple Summary:**

The common vetch (*Vicia sativa* L.) is an important legume crop of mixed crop-livestock systems that provides high-quality grains used as food/feed and straw used as ruminant feed. The objective of this study was to determine the variability in grain yield, straw yield, straw chemical composition, carbohydrate and protein fractions, in vitro gas production, and in situ ruminal degradability of four different varieties of common vetch grown on the Qinghai-Tibetan Plateau. The results showed that grain yield, straw yield, and straw nutrient value varied significantly among the four varieties. Overall, the findings indicated that in terms of straw yield and nutritive quality, variety Lanjian No. 1 has the greatest potential as a crop for supplementing ruminant diets in the smallholder mixed crop–livestock systems on the Qinghai-Tibetan Plateau.

**Abstract:**

Four varieties of common vetch, including three improved varieties (Lanjian No. 1, Lanjian No. 2, and Lanjian No. 3) and one local variety (333A), were evaluated for varietal variations in grain yield, straw yield and straw quality attributes on the Qinghai-Tibetan Plateau. Crops were harvested at pod maturity to determine grain yield, straw yield, harvest index, and potential utility index (PUI). Straw quality was determined by measuring chemical composition, carbohydrate and protein fractions, in vitro gas production and in situ ruminal degradability. Results showed a significant effect (*p* < 0.01) of variety on the grain yield [875.2–1255 kg dry matter (DM)/ha], straw yield (3154–5556 kg DM/ha), harvest index (15.6–28.7%) and PUI (53.3–63.2%). Variety also had a significant effect on chemical composition, carbohydrate and protein fractions (*p* < 0.05) except non-structural carbohydrates and rapidly degradable sugars. Significant differences (*p* < 0.05) were observed among the varieties in potential gas production [188–234 mL/g DM], in vitro organic matter (OM) digestibility (43.7–54.2% of OM), and metabolizable energy (6.40–7.92 MJ/kg DM) of straw. Significant differences (*p* < 0.001) were also observed among the varieties in rapidly degradable DM fraction and effective DM degradability of straw; however, no difference was observed in other DM degradation parameters and neutral detergent fiber degradation parameters. In conclusion, based on straw yield and quality, Lanjian No. 1 has the greatest potential among the tested varieties as a crop for supplementing ruminant diets for smallholder farmers on the Qinghai-Tibetan plateau.

## 1. Introduction

Ruminants provides the majority of food worldwide, contributing approximately 25.9% of global meat production and nearly 100% of global milk production [1,2]. The demand for cattle and sheep meat has been growing at a global growth rate of 2.0% per year from 2016 and is predicted to continue till 2026 [3]. The Qinghai-Tibetan plateau supports approximately 41 million sheep and over 14.3 million cattle, one of the largest livestock systems in Asia [4,5]. The plateau is also the headwater region of most of Asia’s five major rivers, including Yangtze, Yellow, Mekong, Ganges, and Indus rivers [4]. Thus, sustainable management of the Plateau is important for the livelihood of over 9.8 million nomadic populations and for protecting these crucial river systems.

Unfortunately, harsh climatic conditions (e.g., low temperatures) and short growing season on the Qinghai-Tibetan plateau limit crop production [4,5,6]. Combined with grassland degradation, the annual fodder gap in this region is expected to reach 2 million tonnes, which implies that each year 2.7 million sheep units with insufficient feedstuffs will result in 20–30% liveweight loss during winter and early spring [5,7]. An effective remedy to stabilize feed supply is to incorporate nitrogen-fixing annual cool-season feed legumes such as common vetch (*Vicia sativa* L.) into fallow lands that will improve soil properties and lead to increased grain and straw production [8].

The use of common vetch in the mixed crop-livestock systems has expanded greatly in the last decade [4,9,10,11]. Several studies showed high generation of crop residues after harvesting common vetch that has substantial potential to be used for feeding ruminant livestock [9,12]. Previous studies also reported that the nutritive quality of common vetch straw is relatively high, containing an average of 9.41% crude protein (CP), 55.2% in vitro organic matter digestibility (IVOMD) and 7.3 MJ/kg metabolizable energy (ME) [9,13]. This makes common vetch straw a good feed supplement for ruminants, which are offered low-quality cereal straw-based diets in many small-holder crop-livestock systems [9,14]. In addition, the fatty acid composition of grains has increased the use of common vetch as a feed for ruminants [15]. Although common vetch is important as a feed for ruminants, limited information is available on the varietal variations in straw composition (carbohydrate and protein fractions), in vitro gas production, and in situ ruminal degradability, because most of the earlier studies focused on grain yield, straw yield, and straw quality including only chemical composition, IVOMD and ME [9,10,11,12,13]. Russel et al. [16] suggested that carbohydrate and protein fractions can be used as a reliable indicator to accurately predict biological value and performance of feed in ruminants. Marcos et al. [17], and Blümmel and Ørskov [18] demonstrated that the chemical composition in combination with in vitro digestibility and in situ ruminal degradability can be used as crucial parameters to evaluate the nutritive value of feed. 

Accordingly, the objective of this study was to evaluate grain and straw yields, as well as chemical composition, carbohydrate and protein fractions, in vitro gas production and in situ ruminal degradability of straw of four different common vetch varieties grown on the Qinghai-Tibetan Plateau.

## 2. Materials and Methods 

### 2.1. Location, Experimental Design, and Sampling

A detailed description of the location, experimental design and sampling is in the companion paper on the nutritive value of common vetch grain [4] and are summarized here.

This experiment was conducted during the cropping season of 2015 at the Xiahe Experimental Station of Lanzhou University, Gansu, China (35°45′ N, 102°34′ E; altitude 2880 m). The region under this study is situated at the eastern margin of the Qinghai-Tibetan Plateau. The location received an annual average precipitation of 452 mm (80% in May and September) and recorded a mean annual air temperature of 3.5 °C (1984–2014, 31 years). The previous crop was rape (*Brassica campestris* L.).

Three improved varieties (Lanjian No. 1, Lanjian No. 2, Lanjian No. 3) and one local variety (333A) were utilized in the field study. These improved varieties, developed by the Common Vetch Breeding Program of Lanzhou University, are well-adapted and extensively grown by smallholder farmers in the region surrounding the research site. Agronomic characteristics of these tested varieties are shown in Table 1. All four varieties were planted under the same agronomic conditions following a completely randomized design with four replicates. The individual plot area was 40 m^2^ (8 m × 5 m) with a row spacing of 20 cm, and grains were sown by hand at a depth of 3–5 cm at a density of 150 viable grains m-2. Grains were sown (6 May 2015) before inoculation with rhizobium (CCBAU01069, China Agricultural University, Beijing, China), which was recommended based on the symbiont performance of these varieties [19]. Irrigation and fertilizers were not applied after sowing, and weeds in each plot were adequately controlled manually.

Harvesting of the common vetch plants was done at the pod maturity (6, 15, 15, and 26 September for Lanjian No. 3, 333A, Lanjian No. 2 and Lanjian No. 1, respectively). The crops were manually harvested from two representative subplots (1 m × 1 m) of each plot for each variety and threshed to obtain grain and straw samples. Harvest index was calculated as follows: Harvest index (%) = Grain yield × 100 / (Grain yield + Straw yield). All samples were oven-dried at 65 °C for 48 h and ground to pass through a 2-mm sieve for analysis of in situ ruminal incubation and to pass through a 1-mm sieve for chemical analysis and in vitro gas production measurement. All procedures involved animals were approved by the Animal Ethics Committee of Lanzhou University (protocol AEC-LZU-2016-01). 

### 2.2. In Vitro Gas Production

Four adult Dorper rams (approximately 33-month-old; 58.4 ± 1.24 kg body weight) fitted with flexible rumen cannulas were used as donors of ruminal fluid. The rams were kept in individual stalls and had free access to fresh water and mineral/vitamin licks. The rams were daily fed 1.2 kg of 550 g/kg DM sheepgrass [*Leymus chinensis*, (Trin.) Tzvel], 140 g/kg DM soybean meal, 294 g/kg DM maize (*Zea mays* L.) seed, 8.6 g/kg DM calcium hydrophosphate, 5.0 g/kg DM salt and 2.4 g/kg DM mineral-vitamin mix at maintenance energy level in equal portions at 08:00 and 16:30 hours. In vitro gas production (GP) was measured as described by Blümmel and Ørskov [18]. Rumen fluids were obtained before morning feeding and strained through four layers of cheesecloth into a preheated, insulated bottle. Briefly, approximately 200 mg DM of each sample (in duplicate) was weighed into calibrated glass syringes (100 mL). Each syringe was preheated at 39 °C before injecting 30 mL rumen fluid/buffer mixture [20]. Then, the syringes were placed vertically in a water bath at 39 °C with three syringes without a sample used as blank. The volume of GP was manually recorded after 0, 4, 6, 8, 12, 24, 36, 48 and 72 h of incubation and was blank corrected. The incubation run was repeated three times. All operations involving rumen fluid were conducted under a continuous flush of CO_2_ to ensure anaerobic conditions. 

The GP data were fitted with time using the following equation [17], as follows: *y* = *b* × [1 – e^−*c* (*t* - *lag*)^], where *y* = the volume of GP at time *t*; *b* = the potential GP (mL/g DM); *c* (h^−1^) = the fractional rate of GP (h^−1^); *lag* = the initial delay in the onset of gas production (h); *t* = incubation time (h). Parameters *b*, c and *lag* were determined by an iterative least square method using the NLIN procedure of SAS 9.2 (SAS Inst. Inc., Cary, NC). The IVOMD [% organic matter (OM)] and ME (MJ/kg DM) were estimated using the equations of Menke and Steingass [21] as: IVOMD = 14.88 + 0.889 GP + 0.45 CP + 0.0651 ASH; ME = 2.20 + 0.136 GP + 0.057 CP + 0.0029 EE^2^, where GP = the net gas production after 24 h of incubation (mL/200mg DM), ASH = the ash content (% DM), CP = the crude protein content (% DM), and EE = the ether extract content (% DM).

### 2.3. In Situ Ruminal Incubation

In situ ruminal degradability of DM and neutral detergent fiber (NDF) of straw samples was determined using the nylon bag technique described by Nandra et al. [22]. Briefly, approximately 5.0 g of each sample (in duplicate) was weighted into nylon bags (9 × 5 cm; 50-μm pore size) and incubated in the ventral sacs of the rumen of the same four rams used for the production of ruminal fluid in Section 2.2. The bags were inserted into the rumen for 0 (control), 4, 8, 12, 24, 48 and 72 h. Following removal, the bags were briefly washed under cold running water and frozen (–20°C) until further analysis. All bags were defrosted, manually washed in cold tap water until the water was clear, oven-dried at 60℃ for 48 h, and weighed. The dried undigested residues of replicates per same time within sheep were pooled to measure DM and NDF. Straw DM and NDF disappearance rates were estimated from the difference in straw weight before and after incubation. The kinetic parameters of DM and NDF degradation were determined using the exponential equation described by Ørskov and McDonald [23]. The effective degradability of DM (EDDM) and NDF (EDNDF) were calculated as ED= A + [(B × C)/(C + k)], where A = the soluble fraction, B = the potentially degradable fraction, C = the rate of degradation of fraction B, k = the rumen outflow rate (0.031 h^−1^) [4].

### 2.4. Laboratory Analysis

Determination of DM (ID 930.15), nitrogen (N; ID 988.05), ether extract (EE; ID 920.85), ash (ID 938.08), acid detergent fiber (ADF; ID 973.18) and acid detergent lignin (ADL; ID 973.18) were analyzed following the methods of the Association of Official Agricultural Chemists [24]. The CP content was calculated by multiplying the nitrogen value by 6.25. The NDF content was determined following the method by Van Soest et al. [25] using heat-stable α-amylase and sodium sulfite. Contents of NDF and ADF were expressed inclusive of residual ash. Acid detergent insoluble protein (ADIP), neutral detergent insoluble protein (NDIP) were measured by Kjeldahl analysis of the ADF and NDF bag residues, respectively, using the procedure described by Licitra et al. [26].

Carbohydrate fractions of straw samples from the four common vetch varieties were determined as proposed by the Cornell Net Carbohydrate and Protein system (CNCPS) [27]. The system divides carbohydrate into four fractions in terms of their degradation rate as follow: C_A,_ rapidly degradable sugars; CB_1,_ intermediately degradable pectin and starch; CB_2,_ slowly degradable cell wall; and C_C,_ undegradable/lignin-bound cell wall. Total carbohydrates (TCHO) content was calculated as TCHO = 100 – (CP + EE + Ash) [28]. Non-structural carbohydrates (NSC) and structural carbohydrates (SC) were estimated using the equations given by Caballero et al. [29] as: NSC = TCHO – SC and SC = NDF – NDIP. Starch content was determined by enzymatic hydrolysis of α-linked glucose polymers [30].

The CP of straw samples was fractionated into five different fractions according to CNCPS as described by Licitra et al. [26] and Sniffen et al. [27]. These fractions include: fraction P_A_, non-protein nitrogen (NPN), calculated as the difference between total N and true protein N, analyzed using sulphuric acid (0.5 M) and sodium tungstate (0.30 M); fraction P_B1_, buffer-soluble protein, estimated by subtracting buffer-insoluble protein precipitated with freshly prepared (1 g/10 mL) sodium azide and borate-phosphate buffer (pH 6.7–6.8) solution from true protein; fraction P_B2_, neutral detergent- soluble protein, calculated by subtracting NDIP from buffer-insoluble protein; fraction P_B3_, acid detergent-soluble protein, calculated by subtracting ADIP from NDIP; and fraction P_C_, ADIP, is indigestible protein. All measurements were performed in duplicate and appropriate chemical standards were included in each analytical run. 

### 2.5. Calculations and Statistical Analyses

Potential utility index (PUI) was estimated from the amount of utilizable portion of the total biomass yield for grain and straw regardless of the economic value as described by Alkhtib et al. [31]:PUI (%) = 100 × [grain yield + 0.01 × IVOMD (%) × straw yield]/ total biomass yield

Data collected were subjected to one-way ANOVA using SPSS software (Version 21.0. IBM Corporation, Armonk, NY, USA). The fermentation parameters were subjected to separate analysis of variance with varieties as fixed effect and the incubation run as random effect. Differences between means were compared using the Duncan significant difference test at *p* < 0.05.

## 3. Results

### 3.1. Grain Yield, Straw Yield, and PUI 

Table 2 shows a significant effect of variety on grain yield and straw yield (*p* < 0.01). The grain yield ranged from 875.2 to 1255 kg DM/ha with an average value of 1088 kg DM/ha. The grain yield of 333A and Lanjian No. 1 was significantly less than that of other improved varieties. The straw yield varied from 3154 to 5556 kg DM/ha. The least straw yield was observed for Lanjian No. 3 and the greatest for Lanjian No. 1. 

Variety had a significant (*p* < 0.001) effect on harvest index and PUI. Harvest index varied from 15.6 to 28.7%. Harvest index of Lanjian No. 2 was less than that of Lanjian No. 3, but greater than that of the other varieties (*p* < 0.01). The PUI varied from 53.3 to 63.2%. The PUI of the local variety was less than the improved varieties, which had similar PUI (average 62.1%). 

### 3.2. Chemical Composition

Table 3 shows a significant effect of variety on DM, ash, CP and EE contents of the straw samples (*p* < 0.01). The DM content among varieties varied from 90.1 to 90.5% and was the greatest in Lanjian No. 2. The ash content varied from 10.2 to 13.5% DM. The least ash content was for 333A and the greatest for Lanjian No. 1. The CP content varied from 9.76 to 13.8% DM. The local variety had significantly less CP content compared to the improved varieties. There were significant differences in CP content among the improved varieties, which was greater for Lanjian No. 1. The EE content varied from 0.459 to 1.11% DM. The EE content of 333A was similar to that of Lanjian No. 3 but was significantly greater than that of Lanjian No. 1 and Lanjian No. 2. 

Significant differences (*p* < 0.05) were observed in the cell wall contents of different varieties. The NDF, ADF, and ADL contents varied from 45.0 to 54.1% DM, 27.4 to 33.2% DM, and 6.08 to 9.56% DM, respectively. The NDF, ADF and ADL contents were the greatest in 333A and the least in Lanjian No. 1. Hemicellulose and cellulose contents varied from 17.6 to 21.4% DM and 21.4 to 23.7% DM, respectively. The hemicellulose and cellulose contents of variety 333A were higher than those of Lanjian No. 1, but similar to those of other varieties. Phosphorus content varied from 0.185 to 0.296% DM. The phosphorus content was considerably greater (*p* < 0.05) in Lanjian No. 3 than in Lanjian No. 2 and 333A, with Lanjian No. 1 being intermediate. Calcium content was also significantly influenced by variety (*p* < 0.001). It varied from 1.00 to 1.54% DM and was the greatest in Lanjian No. 1. 

### 3.3. Carbohydrate and Protein Fractions

As shown in Table 4, the TCHO content was significantly different (*p* < 0.001) among varieties and varied from 72.2 to 79.0% DM. The greatest TCHO content was observed in 333A and the least in Lanjian No. 1. The NSC content varied from 29.3 to 31.2% DM with no difference (*p* > 0.05) among varieties. The SC content varied from 41.1 to 49.7% DM. The variation in SC content was the greatest in 333A and the least in Lanjian No. 1. Significant differences (*p* < 0.05) were observed in CHO fractions except C_A_ fraction (Table 4). The C_B1_ fraction was greatest (*p* < 0.001) in Lanjian No. 1 (26.0% CHO), intermediate in Lanjian No. 2 (22.9% CHO), and least in Lanjian No. 3 (20.1% CHO) and 333A (17.7% CHO). The C_B2_ fraction varied from 33.9 to 39.5% CHO. Variety 333A had significantly less (*p* < 0.05) C_B2_ fraction than Lanjian No. 2 and Lanjian No. 3 but similar to Lanjian No. 1. The Cc fraction varied from 20.2 to 29.0% CHO. The C_C_ fraction of variety 333A was significantly greater (*p* < 0.001) compared to the improved varieties, which had similar C_C_ fraction.

Variety had a significant influence on the protein fractions of straw (*p* < 0.01; Table 4). The P_A_ fraction varied from 7.16 to 11.5% CP, and the least value was recorded in 333A and the greatest in Lanjian No. 1. The P_B1_ fraction varied from 25.8 to 37.3% CP; the greatest value was recorded in 333A. The P_B2_ fraction varied from 10.8 to 34.0% CP. The variation in P_B2_ fraction was the least in 333A and the greatest in Lanjian No. 1. The P_B3_ fraction varied from 13.3 to 23.6% CP. The P_B3_ fraction was significantly greater for 333A than for Lanjian No. 3, and significantly greater for Lanjian No. 3 than the other varieties. The P_C_ fraction varied from 15.4 to 21.1% CP. The P_c_ fraction was greatest in 333A and least in Lanjian No. 1. 

### 3.4. In Vitro Gas Production

Table 5 shows the gas production parameters, IVOMD and ME of straw in four common vetch varieties. The potential gas production differed considerably (*p* < 0.001) between the varieties and varied from 188 to 234 mL/g DM. The potential gas production of Lanjian No. 3 was greater than that of 333A; however, it was less than that of the other varieties. The fractional rate of GP was not affected by variety (*p* > 0.05), averaging 0.0631 h^−1^. The *lag* value was similar (*p* > 0.05) for the four varieties (average 0.633 h).

Variety had a significant influence on the IVOMD content of straw (*p* < 0.001; Table 5). Straw IVOMD varied from 43.7 to 54.2% OM, and the varieties were ranked in order Lanjian No. 1 > Lanjian No. 2 > Lanjian No. 3 > 333A. Significant difference in the ME content was also observed among the different varieties (*p* < 0.001), and it varied from 6.40 to 7.92 MJ/kg DM. The least value was recorded for 333A and the greatest for Lanjian No. 1 and Lanjian No. 2.

### 3.5. In Situ Ruminal Degradability

The soluble DM fraction was significantly different (*p* < 0.001) among varieties and varied from 22.8 to 28.4% DM (Table 6). The soluble DM fraction of 333A was similar to that of Lanjian No. 3, but less than that of Lanjian No. 1 and Lanjian No. 2. The potentially degradable DM fraction and rate of DM degradation were similar (*p* > 0.05) for the four varieties and recorded an average of 43.2% DM and 0.0442 h^−1^, respectively. The EDDM value was significantly different (*p* < 0.001) among the four varieties. Average EDDM value was 51.0% DM and ranged from 46.7 to 55.2% DM. Variety 333A had significantly less EDDM value compared to Lanjian No. 1 and Lanjian No. 2, but it was similar to Lanjian No. 3. 

The NDF degradation profiles in situ are given in Table 5. Soluble NDF fraction, potentially degradable NDF fraction, and rate of NDF degradation were not influenced (*p* > 0.05) by variety and recorded an average of 11.5% NDF, 46.7% NDF and 0.0374 h^−1^, respectively. No difference was observed in EDNDF value among varieties, but there was a trend toward greater EDNDF for Lanjian No. 1 (*p* = 0.078).

## 4. Discussion 

### 4.1. Grain Yield, Straw Yield, and PUI

In smallholder crop-livestock systems, improvement in crop straw yield implies increase in milk and meat production from ruminants [6,32]. Substantial variability differences among varieties were found for grain and straw yields, partly due to differences in days to pod maturity and harvest index as suggested by Abd El-Moneim [8], Larbi et al. [9], and Kafilzadeh and Maleki [32]. Our findings are consistent with the earlier reports on common vetch [9] as well as faba bean (*Vicia faba* L.) [31], lentil (*Lens culinaris*) [32] and chickpea (*Cicer arietinum*) [33]. The grain yield recorded in this study (875.2–1255 kg DM/ha) is within the reported range (287–1783 kg DM/ha) [9], but less than ranges reported value of 1340–2240 kg DM/ha [12]. Meanwhile, the straw yield (3154–5556 kg DM/ha) is greater than that reported by Larbi et al. [9] (629–2226 kg DM/ha), but slightly less than that reported by Albayrak et al. [12] (4620–7320 kg DM/ha). The yields vary between studies [9,12] as consequence of the differences in the varieties, agronomic practices and growing conditions (e.g., soil type and climate) [8,12,34]. The grain yield of the improved varieties was significantly greater than that of the local variety in this study. Similar results were reported for other leguminous crops such as faba bean [31] and lentil [35]. The local variety demonstrated inferior PUI compared to the improved varieties. This is consistent with the findings in faba bean [31]; however, contrary to the findings of Tolera et al. [34] who observed less PUI for the improved varieties compared with the local varieties.

### 4.2. Chemical Composition 

Higher CP and lower cell wall contents (NDF, ADF, cellulose and ADL) can be used as indicators of good feed quality [28]. Large varietal differences in straw chemical composition observed in our study is in agreement with the earlier findings in common vetch [9,11,13], as well as faba bean [31], lentil [32] and chickpea [33]. The range of CP content observed in this study (9.76–13.8% DM) was more than the threshold content (8.0% DM CP) required for optimum activity of rumen microorganisms in ruminants [36]. The NDF content among the varieties varied from 45.0 to 54.1% DM with a mean value of 49.9% DM. Van Soest [37] indicated that NDF content over 65% leads to effects on voluntary intake and production by ruminants. This makes common vetch straw a good source of CP supplements for ruminants in smallholder crop-livestock/agro-pastoral systems. Common vetch straw NDF and ADF contents of 52.2 and 36.1% DM reported by Makkar et al. [13] are consistent with our results; however, the CP contents (6.2% DM) is less than our results. Phosphorus and calcium contents observed in this study are similar to those reported by Abreu and Bruno-Soares [38]. The differences in chemical composition of straw between studies may be due to varietal variability, differences in growing condition (e.g., soil type and climate), or differences in harvesting and postharvest handling practices.

### 4.3. Carbohydrate and Protein Fractions

The significant varietal differences in carbohydrate fractions of common vetch straw are in agreement with reported for sorghum (*Sorghum bicolor* [L.] Moench) [28], timothy (*Phleum pratense* L.) [39] and wheat (*Triticum aestivum* L.) [40]. The values for TCHO, NSC and SC obtained in this study are comparable with the previous studies on other legume crops such as *Trifolium alaxendrinum* [41]. Among the CHO fractions, C_B2_ fraction was the highest in the straw of common vetch varieties analyzed. Others have reported similar results in sorghum, berseem (*Trifolium alexandrium*) and cowpea (*Vigna sinensis*) [28,41]. However, there is limited data available on the CHO fractions of common vetch straw. The pattern of CHO fractions observed in our study for common vetch straw is comparable with the earlier reports on other forage crops [41,42,43].

Wide range of protein fractions observed in the varieties is consistent with the findings in sorghum [28] and wheat [40]. Swarna et al. [44] indicated that P_B2_ and P_B3_ fractions represent a bypass protein of forage, while P_c_ fraction represents the non-degraded fraction. Compared to other varieties, Lanjian No. 1 had lower P_c_ fraction and higher P_B2_ + P_B3_ fractions. These observations on protein fractions suggest that straw of variety Lanjian No. 1 could be used as the better nitrogen source for ruminants. There is limited data available on the contents of protein fractions of common vetch. The pattern of protein fractions revealed here is similar to the reports on lucerne (*Medicago sativa* L.) [43] and black gram [*Vigna mungo* (L.) Hepper] [44].

### 4.4. In Vitro Gas Production 

The gas produced by in vitro fermentation reflects the degree of feed fermentation and digestibility [34]. In this study, we observed significant differences among varieties in potential gas production, but not in gas production rate and *lag* time. Our results are partially in line with previous studies on other crops such as spineless cacti (*Opuntia* spp) [20] and chickpea [33]. Studies on gas production from common vetch straws are scarce, with the exception of the Spain studies [45]. López et al. [45] reported greater potential gas production, and less gas production rate and *lag* time in comparison with those observed in the current study. Cone and van Gelder [46] indicated that despite high degradability, feed with high CP typically produce less gas during fermentation as protein fermentation produces ammonia, which affects the carbonate buffer balance by neutralizing H^+^ ions from volatile fatty acids without releasing carbon dioxide. The differences in gas production parameters between studies may be attributed to the differences in straw CP content, varieties, season, and location.

The IVOMD and ME contents of straw varied among the varieties, which is partly due to varietal variations in straw ADF and NDF and may be due to the proportions of straw morphological fractions, which were not measured in this study. In this study, varieties Lanjian No. 1 and Lanjian No. 2 recorded straw CP and IVOMD as high as 8.0% DM and 50.0% OM, respectively, which suggests that the straw of these varieties may be effectively used as a CP supplement to ruminants fed low-quality cereal straw-based diets [9]. The straw IVOMD range in this study is comparable to the range reported earlier in other vetch varieties [9], while the ME range is slightly less than those reported for vetches and other legume straws [13]. The straw quality varied between studies possibly due to differences in varieties, cell wall lignification, leaf to stem ratio, and the stage at which the straw was harvested. Larbi et al. [9] and Makkar et al. [13] earlier reported that straw IVOMD and ME contents of common vetch are influenced by varieties, growing season, and stage of straw harvest. 

### 4.5. In Situ Ruminal Degradability

Higher ruminal degradability of high-fiber forages is satisfying because it implies improved the nutrient availability to rumen microbes [47]. The observed differences in DM degradation profiles of the straw varieties may be related to their varietal traits reflected as substantial differences in morphological and chemical composition. Our results are consistent with those reported for other crops such as maize [47]. For DM degradation parameters, the value of *A* fraction observed in this study for common vetch straw is greater and the values of *B* and *C* fractions are similar to those reported by Bruno-Soares et al. [48]. In this study, the value of *A* fraction of Lanjian No. 1 was significantly greater than other varieties, while the values for *B* and *C* fractions were similar for the four varieties analyzed. The differences in EDDM between the varieties can be mainly attributed to the differences in *A* fraction and not to *B* and *C* fractions, which is consistent with the literature [47]. The EDDM reported for common vetch straw in our study are similar to that reported for chickpea straw (51.8% DM) [49], but greater than that reported for fenugreek (*Trigonella foenum-graecum*) straw (32.2% DM) [50]. The degradation profiles of NDF were not influenced by variety in our study, which is in agreement with the earlier reports on maize and chickpea [47,49]. Bruno-Soares et al. [48] reported less rapidly degradable NDF fraction and rate of NDF degradation, and similar potentially degradable NDF fraction in comparison with those observed in this study. The EDNDF of straw in this study was less than reported by Abbeddou et al. [51] for lentil straw (45.9% NDF), but greater than that reported by Mustafa et al. [50] for fenugreek straw (20.3% NDF). Different studies recorded different ruminal degradation kinetics due to differences in varieties/species, straw composition, and animal species [4,49,50,51].

## 5. Conclusions

The results of this study showed varietal differences in grain and straw yields, and straw nutrient value in common vetch. Evaluation of common vetch varieties showed that Lanjian No. 1 had straw yield, straw CP, non-protein nitrogen, neutral detergent soluble protein, IVOMD, ME, and EDDM greater than other varieties, despite its less grain yield. Variety Lanjian No. 3 demonstrated early maturity and greater grain yield; however, straw yield and quality were less than Lanjian No. 1. Variety Lanjian No. 2 had greater grain yield and moderate straw CP; however, straw IVOMD, ME, EDDM and EDNDF contents were comparable with Lanjian No. 1. Based on these results, variety Lanjian No. 1 is the best option among varieties examined for smallholder farmers on the Qinghai-Tibetan plateau.

## Figures and Tables

**Table 1 animals-09-00505-t001:** Agronomic characteristics of the common vetch varieties utilized in this study.

Agronomic Characteristic	333A	Lanjian No.1	Lanjian No.2	Lanjian No.3
Days to mature (day)	134	145	132	124
1000 grains weight (g)	54	79	71	76
Plant height (cm)	92	106	80	69
Altitude (m.a.s.l)	–	<3000	<3500	<4000
Year of release	1987	2014	2015	2011

Source: Ministry of Agriculture, Beijing, China.

**Table 2 animals-09-00505-t002:** Influence of variety on grain yield, straw yield, harvest index, and potential utility index (PUI) of four common vetch varieties.

Dependent Variable	333A	Lanjian No. 1	Lanjian No. 2	Lanjian No. 3	SEM ^1^	*p*-Value
Grain yield (kg DM/ha)	875.2 ^b^	1024 ^b^	1196 ^a^	1255 ^a^	45.1	0.001
Straw yield (kg DM/ha)	4319 ^b^	5556 ^a^	4158 ^bc^	3154 ^c^	267.6	0.003
Harvest index (%)	17.0 ^c^	15.6 ^c^	22.6 ^b^	28.7 ^a^	1.39	<0.001
PUI (%)	53.3 ^b^	61.4 ^a^	62.8 ^a^	63.2 ^a^	0.660	<0.001

^a,b^ Within a raw, different letters represent the significant differences at *p*-value < 0.05. ^1^ DM, dry matter; SEM, standard error of the mean.

**Table 3 animals-09-00505-t003:** Influence of variety on chemical composition (% dry matter unless stated otherwise) of straw in four common vetch varieties.

Dependent Variable	333A	Lanjian No. 1	Lanjian No. 2	Lanjian No. 3	SEM ^1^	*p*-Value
Dry matter %	90.2 ^bc^	90.4 ^ab^	90.5 ^a^	90.1 ^c^	0.055	0.009
Ash	10.2 ^c^	13.5 ^a^	11.9 ^b^	11.0 ^bc^	0.393	0.003
Crude protein	9.76 ^c^	13.8 ^a^	12.4 ^b^	11.5 ^b^	0.424	<0.001
Ether extract	1.06 ^a^	0.459 ^b^	0.515 ^b^	1.11 ^a^	0.088	<0.001
Neutral detergent fiber	54.1 ^a^	45.0 ^c^	49.4 ^b^	51.0 ^b^	0.915	<0.001
Acid detergent fiber	33.2 ^a^	27.4 ^c^	30.3 ^b^	29.6 ^b^	0.603	<0.001
Acid detergent lignin	9.56 ^a^	6.08 ^c^	6.53 ^bc^	7.05 ^b^	0.369	<0.001
Hemicellulose	20.9 ^a^	17.6 ^b^	19.2 ^ab^	21.4 ^a^	0.537	0.025
Cellulose	23.6 ^a^	21.4 ^b^	23.7 ^a^	22.5 ^ab^	0.353	0.039
Phosphorus	0.185 ^b^	0.249 ^ab^	0.206 ^b^	0.296 ^a^	0.0149	0.019
Calcium	1.13 ^bc^	1.54 ^a^	1.00 ^c^	1.32 ^ab^	0.061	<0.001

^a,b^ Within a raw, different letters represent the significant differences at *p*-value < 0.05. ^1^ SEM, standard error of the mean.

**Table 4 animals-09-00505-t004:** Influence of variety on carbohydrate and protein fractions of straw in four common vetch varieties.

Dependent Variable	333A	Lanjian No. 1	Lanjian No. 2	Lanjian No. 3	SEM ^1^	*p*-Value
Carbohydrates (% dry matter)						
TCHO	79.0 ^a^	72.2 ^c^	75.2 ^b^	76.4 ^b^	0.665	<0.001
NSC	29.3	31.2	29.8	29.6	0.474	0.575
SC	49.7 ^a^	41.1 ^c^	45.4 ^b^	46.7 ^b^	0.886	<0.001
Carbohydrate fractions (% CHO)						
C_A_	19.4	17.1	18.1	18.7	0.645	0.435
C_B1_	17.7 ^c^	26.0 ^a^	22.9 ^b^	20.1 ^c^	0.885	<0.001
C_B2_	33.9 ^b^	36.7 ^ab^	39.5 ^a^	39.0 ^a^	0.835	0.042
C_C_	29.0 ^a^	20.2 ^b^	20.8 ^b^	22.1 ^b^	0.998	<0.001
Protein fractions (% CP)						
P_A_	7.16 ^c^	11.5 ^a^	9.43 ^b^	8.55 ^bc^	0.499	0.004
P_B1_	37.3 ^a^	25.8 ^c^	30.1 ^bc^	31.8 ^b^	1.31	0.004
P_B2_	10.8 ^c^	34.0 ^a^	27.9 ^b^	22.3 ^b^	2.36	<0.001
P_B3_	23.6 ^a^	13.3 ^c^	14.8 ^c^	18.4 ^b^	1.10	<0.001
P_C_	21.1 ^a^	15.4 ^c^	17.8 ^b^	18.8 ^b^	0.628	0.001

^a,b^ Within a raw, different letters represent the significant differences at *p*-value < 0.05. ^1^ C_A_, rapidly degradable sugars; C_B1_, intermediately degradable pectin and starch; C_B2_, slowly degradable cell wall; C_C_, unavailable/lignin bound cell wall; CP, crude protein; TCHO, total carbohydrates; NSC, non-structural carbohydrates; P_A_, non-protein nitrogen; P_B1_, buffer soluble protein; P_B2_, neutral detergent soluble protein; P_B3_, acid detergent soluble protein; P_C_, indigestible protein; SC, structural carbohydrates; SEM, standard error of the mean.

**Table 5 animals-09-00505-t005:** Influence of variety on gas production parameters, in vitro organic matter digestibility (IVOMD) and metabolizable energy (ME) of straw in four common vetch varieties.

Dependent Variable	333A	Lanjian No. 1	Lanjian No. 2	Lanjian No. 3	SEM ^1^	*p*-Value
*b* (mL/g DM)	188 ^c^	234 ^a^	228 ^a^	208 ^b^	3.39	<0.001
*c* (h^−1^)	0.0608	0.0650	0.0649	0.0617	0.000740	0.334
*lag* (h)	0.608	0.656	0.656	0.612	0.00981	0.131
IVOMD (% OM)	43.7 ^d^	54.2 ^a^	52.0 ^b^	48.4 ^c^	0.676	<0.001
ME (MJ/kg DM)	6.40 ^c^	7.92 ^a^	7.61 ^a^	7.08 ^b^	0.0995	<0.001

^a,b^ Within a raw, different letters represent the significant differences at *p*-value < 0.05. ^1^
*b*, potential gas production; *c*, fractional rate of gas production; DM, dry matter; *lag*, initial delay in the onset of gas production; OM, organic matter; SEM, standard error of the mean.

**Table 6 animals-09-00505-t006:** Influence of variety on in situ ruminal degradation kinetics of dry matter (DM) and neutral detergent fiber (NDF) of straw in four common vetch varieties.

Dependent Variable	333A	Lanjian No. 1	Lanjian No. 2	Lanjian No. 3	SEM ^1^	*p*-Value
DM						
*A* (% DM)	22.8 ^b^	28.4 ^a^	26.8 ^a^	24.4 ^b^	0.630	<0.001
*B* (% DM)	41.1	45.1	43.9	42.6	0.581	0.066
*C* (h^−1^)	0.0430	0.0453	0.0444	0.0442	0.0005	0.377
EDDM (% DM)	46.7 ^c^	55.2 ^a^	52.7 ^ab^	49.5 ^bc^	0.953	<0.001
NDF						
*A* (% NDF)	10.6	12.2	12.0	11.2	0.535	0.735
*B* (% NDF)	45.3	48.8	46.7	46.0	0.536	0.101
*C* (h^−1^)	0.0366	0.0388	0.0371	0.0373	0.0004	0.142
EDNDF (% NDF)	35.1	39.3	37.4	36.2	0.618	0.078

^a,b^ Within a raw, different letters represent the significant differences at *p*-value < 0.05. ^1^
*A*, soluble fraction; *B*, potentially degradable fraction; *C*, rate of degradation of fraction B; EDDM, effective dry matter degradability; EDNDF, effective neutral detergent fiber degradability; SEM, standard error of the mean.

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
