# Peer review of "Comparative Grain Yield, Straw Yield, Chemical Composition, Carbohydrate and Protein Fractions, In Vitro Digestibility and Rumen Degradability of Four Common Vetch Varieties Grown on the Qinghai-Tibetan Plateau"

_animals, 2019, doi:10.3390/ani9080505_

Round 1
Reviewer 1 Report
All requested comments have been addressed
Author Response
Response to Reviewer 1 Comments
Point 1:English language and style—Extensive editing of English language and style required
Response 1: We carried out a minor revision of the manuscript to improve English and readability using track changes, based on suggestions from an English mother-tongue.
Special thanks to you for your good comments. We hope that the correction will meet with approval.
Reviewer 2 Report
Manuscript ID: animals-556108
Title: Comparative Grain Yield, Straw Yield, Chemical Composition, Carbohydrate and Protein Fractions, In Vitro Digestibility and Rumen Degradability of Four Common Vetch Varieties Grown on the Qinghai-Tibetan Plateau
Comments and Suggestions for Authors
Line 126: Uniform always the unit of measure. Do not use mL g-1 but mL/g. Please correct. Line 132: Uniform always the unit of measure. Do not use mL 200mg-1 but mL/200mg. Please correct.
Lines 205-206: The following sentence: “The local variety had significantly greater CP content compared to the improved varieties” is not true, because the CP content was 9.76% in the variety 333A. Please rewrite or delete this sentence.
Line 214: “The hemicellulose and cellulose contents of variety 333A were higher than those of…” instead of “The hemicellulose and cellulose contents of variety 333A were more than those of….”.
Line 243: “The Pc fraction was greatest in 333A….” instead of “The PB3 fraction was greatest in 333A….”.
Line 256: Uniform always the unit of measure. Do not use mL g-1 but mL/g. Please correct.
Line 258: “… averaging 0.0631 h-1.” instead of “…..averaging 0.0631% h-1”.
Table 5: Uniform always the unit of measure. Do not use mL g-1 but mL/g. Please correct.
Line 399: The following sentence: “…EDNDF compared to other varieties, despite its less grain yield.” is not statistically correct, because EDNDF did not differ between varieties. Please rewrite this sentence.
Author Response
Response to Reviewer 2 Comments
Point 1:English language and style-English language and style are fine/minor spell check required
Response 1: As suggested by the reviewer, we carried out a minor revision of the manuscript to improve English and readability using track changes, based on suggestions from an English mother-tongue.
Point 2:Line 126: Uniform always the unit of measure. Do not use mL g-1 but mL/g. Please correct.
Response 2: As suggested by the reviewer, we changed “mL g-1” to “mL/g” within line 129 of the revised manuscript.
Point 3:Line 132: Uniform always the unit of measure. Do not use mL 200mg-1 but mL/200mg. Please correct.
Response 3: As suggested by the reviewer, we changed “mL 200mg-1” to “mL/200mg” within line 135 of the revised manuscript.
Point 4:Lines 205-206: The following sentence: “The local variety had significantly greater CP content compared to the improved varieties” is not true, because the CP content was 9.76% in the variety 333A. Please rewrite or delete this sentence.
Response 4: As suggested by the reviewer, we changed “The local variety had significantly greater CP content compared to the improved varieties” to “The local variety had significantly less CP content compared to the improved varieties” within lines 209-210 of the revised manuscript.
Point 5:Line 214: “The hemicellulose and cellulose contents of variety 333A were higher than those of…” instead of “The hemicellulose and cellulose contents of variety 333A were more than those of….”.
Response 5: As suggested by the reviewer, we changed “The hemicellulose and cellulose contents of variety 333A were more than those of” to “The hemicellulose and cellulose contents of variety 333A were higher than those of” within lines 218-2019 of the revised manuscript.
Point 6:Line 243: “The Pc fraction was greatest in 333A….” instead of “The PB3 fraction was greatest in 333A….”.
Response 6: As suggested by the reviewer, we changed “The PB3 fraction was greatest in 333A” to “The Pc fraction was greatest in 333A” within lines 246-247 of the revised manuscript.
Point 7:Line 256: Uniform always the unit of measure. Do not use mL g-1 but mL/g. Please correct.
Response 7: As suggested by the reviewer, we changed “mL g-1” to “mL/g” within line 259 of the revised manuscript.
Point 8:Line 258: “… averaging 0.0631 h-1.” instead of “…..averaging 0.0631% h-1”.
Response 8: As suggested by the reviewer, we changed “averaging 0.0631% h-1” to “averaging 0.0631 h-1” within line 261 of the revised manuscript.
Point 9:Table 5: Uniform always the unit of measure. Do not use mL g-1 but mL/g. Please correct.
Response 9: As suggested by the reviewer, we changed “ mL g-1” to “mL/g” within Table 5 of the revised manuscript.
Point 10:Line 399: The following sentence: “…EDNDF compared to other varieties, despite its less grain yield.” is not statistically correct, because EDNDF did not differ between varieties. Please rewrite this sentence.
Response 10: As suggested by the reviewer, we changed “ Lanjian No. 1 had more straw yield, straw CP, non-protein nitrogen, neutral detergent soluble protein, IVOMD, ME, EDDM, and EDNDF compared to other varieties, despite its less grain yield” to “Lanjian No. 1 had straw yield, straw CP, non-protein nitrogen, neutral detergent soluble protein, IVOMD, ME, and EDDM greater than other varieties, despite its less grain yield.” within lines 400-402 of the revised manuscript.
Special thanks to you for your good comments. We hope that the correction will meet with approval.
This manuscript is a resubmission of an earlier submission. The following is a list of the peer review reports and author responses from that submission.
Round 1
Reviewer 1 Report
This work is interesting for for the type of analysis carried out and the place of investigation.
Title - Add "seed yield"
Abstract - Add the name of the local and the 3 improved varieties of vetch.
Line 21: add (Vica sativa L.) after "common vetch"
Introduction - Also the fatty acid composition of seeds has increased the use of the common vetch as a feedstuff for ruminants. See: Renna M., Gasmi-Boubaker A., Lussiana C., Battaglini L.M., Belfayez K., Fortina R. (2014) - Fatty acid composition of the seed oil of selected Vicia L. taxa fromTunisia. Italian Journalof Animal Science 13 (2): 308-316. doi: 10.4081/ijas.2014.3193
Line 73: nutritive value OF feedstuffs
Materials and Methods
More information on the local type and the improved varieties of vetch would help to better understand the different seed and straw yields, and the chemical composition and degradability of straw.
Describe what is "Harvest index (%)" and how is calculated
Line 86: describe the main phenological and morphological characteristics/differences of the local and the improved varieties of vetch.
Line86-87: name of the local variety?
Line 92: check English
Line 97: is "333A" the local variety?
Line 105: Dorper (upper case)
Line 109: add ingredients of the concentrate mix
Line 147: specify if fiber fractions are corrected for residual ash on not
Line 150-157: check subscript of CHO fractions
Results
All Tables: delete "Mean" and add "P-value" column and comments in the text
Line 180: the seed yield of "both" local variety and Lanjian n°1 were significantly lower than other improved varieties
Table 1: what is the meaning of ranking the varieties according to the PUI value? Delete if not necessary
Table 2: add the dry matter (DM) content (%) of the varieties
Table 3: check subscript in "Lanjian n°2"
Discussion
Line 350: Medicago sativa L.
Line 351: Vigna mungo
Conclusions
The conclusions are excessively concise, and suggest the use of a single variety of vetch (Lanjian n°1) for its straw yield and chemical composition and characteristics. Please add considerations on the possibility of using the other improved varieties (or the local variety) with higher seed yield in specific agronomic or environmental situations.
Author Response
Response to Reviewer 1 Comments
Point 1:Title - Add "seed yield"Response 1: As suggested by the reviewer, we added “Seed Yield” within line 2 of the revised manuscript.
Point 2: Abstract - Add the name of the local and the 3 improved varieties of vetch.
Response 2: As suggested by the reviewer, we added name of the local and the 3 improved varieties of vetch. “333A, Lanjian No.1, Lanjian No.2, and Lanjian No.3” within lines 21-22 of the revised manuscript.
Point 3:Line 21: add (Vicia sativa L.) after "common vetch"
Response 3: As suggested by the reviewer, we added “(Vicia sativa L.): after common vetch within line 21 of the revised manuscript.
Point 4:Introduction - Also the fatty acid composition of seeds has increased the use of the common vetch as a feedstuff for ruminants. See: Renna M., Gasmi-Boubaker A., Lussiana C., Battaglini L.M., Belfayez K., Fortina R. (2014) - Fatty acid composition of the seed oil of selected Vicia L. taxa fromTunisia. Italian Journalof Animal Science 13 (2): 308-316. doi: 10.4081/ijas.2014.3193.
Response 4: As suggested by the reviewer, we added the sentence from the reference “In addition, the fatty acid composition of seeds has increased the use of the common vetch as a feedstuff for ruminants [14]” within lines 65-67 of the revised manuscript.
Point 5: Line 73: nutritive value OF feedstuffs
Response 5: As suggested by the reviewer, we changed “nutritive value from feedstuffs” to “nutritive value of feedstuffs” within line 75 of the revised manuscript.
Point 6: More information on the local type and the improved varieties of vetch would help to better understand the different seed and straw yields, and the chemical composition and degradability of straw.
Response 6: As suggested by the reviewer, we added the sentence “These improved varieties were developed by the Common Vetch Breeding Program of Lanzhou University” and “Agronomic characteristics of these tested are shown in Table 1.” within lines 91-92, 94 of the revised manuscript.
Point 7: Describe what is "Harvest index (%)" and how is calculated
Response 7: As suggested by the reviewer, we added the sentence “Harvest index was calculated as Harvest index (%) = Seed yield × 100 / (Seed yield + Straw yield).” within lines 105-106 of the revised manuscript.
Point 8: Line 86: describe the main phenological and morphological characteristics/differences of the local and the improved varieties of vetch
Response 8: As suggested by the reviewer, we added the sentence “Agronomic characteristics of these tested are shown in Table 1” within lines 94, 111 of the revised manuscript.
Point 9: Line86-87: name of the local variety?
Response 9: As suggested by the reviewer, we added this information “termed 333A” within line 91 of the revised manuscript.
Point 10: Line 92: check English
Response 10: As suggested by the reviewer, we changed “The seeds were hand sown on the 6 May 2015 before being inoculation with Rhizobium leguminosarum (CCBAU01069, China Agricultural University)” to “Crops were planted on the 6 May 2015 following rhizobial inoculation (CCBAU01069, China Agricultural University, Beijing, China)” within lines 97-99 of the revised manuscript.
Point 11: Line 97: is "333A" the local variety?
Response 11: Yes, Thank you so much. As suggested by the reviewer, we added this information “local variety, i.e., 333A” within line 91 of the revised manuscript.
Point 12: Line 105: Dorper (upper case)
Response 12: As suggested by the reviewer, we changed “dorper” to “Dorper” within line 114 of the revised manuscript.
Point 13: Line 109: add ingredients of the concentrate mix
Response 13: As suggested by the reviewer, we changed “16 g/kg DM of a concentrate mix” to “8.6 g/kg DM calcium hydrophosphate, 5.0 g/kg DM salt and 2.4 g/kg DM mineral-vitamin mix” within lines 118-119 of the revised manuscript.
Point 14: Line 147: specify if fiber fractions are corrected for residual ash on not.
Response 14: As suggested by the reviewer, we added the sentence “Contents of NDF, ADF, and ADL were estimated not to be corrected for ash content.” within lines 156-157 of the revised manuscript.
Point 15: All Tables: delete "Mean" and add "P-value" column and comments in the text
Response 15: As suggested by the reviewer, we deleted "Mean" and added "P-value" in all Tables.
Point 16: Line 180: the seed yield of "both" local variety and Lanjian n°1 were significantly lower than other improved varieties
Response 16: As suggested by the reviewer, we changed “Seed yield of the local variety was similar to that of Lanjian No.1, but significantly lower than that of Lanjian No.2 and Lanjian No.3.” to “The seed yield of varieties 333A and Lanjian No.1 was significantly lower than that of other improved varieties.” within lines 192-193 of the revised manuscript.
Point 17: Table 1: what is the meaning of ranking the varieties according to the PUI value? Delete if not necessary
Response 17: As suggested by the reviewer, we deleted “ranking the varieties according to the PUI value” in Table 1.
Point 18: Table 2: add the dry matter (DM) content (%) of the varieties
Response 18: As suggested by the reviewer, we added “DM content” and “The DM content among varieties varied from 90.1 to 90.6 % DM, and was highest in Lanjian No.2.” within lines 206-207 of the revised manuscript.
Point 19: Table 3: check subscript in "Lanjian No. 2"
Response 19: We have made a revision according to referee’s comments.
Point 20: Line 350: Medicago sativa L.
Response 20: As suggested by the reviewer, we changed “Medicago Sativa L.” to “Medicago sativa L.” within line 356 of the revised manuscript.
Point 21: Line 351: Vigna mungo
Response 21: As suggested by the reviewer, we changed “Vignamungo” to “Vigna mungo” within line 357 of the revised manuscript.
Point 22: The conclusions are excessively concise, and suggest the use of a single variety of vetch (Lanjian n°1) for its straw yield and chemical composition and characteristics. Please add considerations on the possibility of using the other improved varieties (or the local variety) with higher seed yield in specific agronomic or environmental situations.
Response 22: As suggested by the reviewer, we added the sentences “Lanjian No.3 had earlier mature and higher seed yield, but lower yield and quality than Lanjian No.1. The Lanjian No.2 had more seed yield and moderate CP, but comparable straw IVOMD, ME, EDDM and EDNDF contents as Lanjian No.1.” within lines 410-412 of the revised manuscript.
Additional minor revisions were made to the manuscript using track changes to improve English and readability.
Special thanks to you for your good comments. We hope that the correction will meet with approval.
Reviewer 2 Report
L79-103: 2.1. Location, experimental design and sampling
It can be shortened by citing Huang et al., 2019, Animal, 9:..
L111 Ørskov [16]. Rumen fluids were obtained before morning feeding from 4 adult fistulated dorper
Delete. It had been described above
L117-118: 6, 8, 12, 24, 36, 48 and 72 h of incubation and corrected for the blanks. The whole process was repeated on a different day. All operations involving rumen fluid were conducted under a…
Only two repetitions. That is an important gap. Most of lab studies are done with three repetitions due to the important factors determining the variability of results (animals, ruminal fluid, moment of assay…). Is it advisable to do always 3 repetitions (3 runs in three different days)
L147 Soest et al. [23] using heat‐stable α‐amylase and sodium sulfite. Acid detergent insoluble protein
heat‐stable α‐amylase is not necessary because straw does not have starch
L173-173:
I’m not statistical expert, but the fermentation kinetics parameters should be done with mixed models, considering the variety as fixed effect and the run as random effect.
Results
Results that are expressed in the Table must not be written in the main text, Please delete all.
L231: Lanjian No.2 (22.9 % CHO), and lowest for Lanjian No.3 (20.1 % CHO) and 333A (17.7% CHO).
333A: is it the local variety??? It has not been described before
L301-305: bean (Vicia faba L.) [29] and chickpea (Cicer arietinum) [31]. The seed yield range (875.2–1255 kg DM/ha) in this study is within ranges reported value of 287–1783 kg DM/ha [8], but lower than ranges reported value of 1340–2240 kg DM/ha [11]. The straw yield range (3154‐5556 kg DM/ha) is higher than reported ranges (629–2226) by Larbi et al. [8], but slightly lower than the range (4620 –
305 7320 kg DM/ha) reported by Albayrak et al. [11]. These
It can be sumarized saying that: yields vary between studies (8, 11...) as consequence of the differences in the varieties, agronomic practices and growing conditions (e.g., soil type and climate)....
L319: 60% DM CP
What does it mean??
L323-327: Results of this study showed that ADF, ADL, hemicellulose and cellulose contents of the straws varied from 27.4 to 33.2 % DM, 6.08 to 9.56 % DM, 17.6 to 21.4 % DM, and 21.4 to 23.7 % DM, respectively. Fortina [10] reported higher NDF, ADF, ADL and cellulose (62.8, 46.7, 9.73 and 37.0 %DM) and similar CP and EE contents (10.1 and 0.63 %DM) to our results.
Can be deleted
L361363: The average values for potential gas production observed in the current study are lower than the values reported by López et al. [45] despite the gas 363 production rate was lower than in the this study
Delete.
L363: The differences
L372-376: Mean ME value in common vetch straw (6.6 MJ/kg DM) recorded by Abreu and Bruno‐Soares [36] were slightly lower than values in the current study, but similar to values observed by Abreu and Bruno‐Soares [36] and López et al. [45]. Such differences in IVOMD and ME content of common vetch straw among studies may be related with natural variation in common vetch straws themselves, as well as plant growing conditions [8,12,43].
No sense. Rewrite the sentence. It is inconsistent with some sentences wrote before
The Tables: Add the column of the p values of effect.
Author Response
Response to Reviewer 2 Comments
Point 1:L79-103: 2.1. Location, experimental design and samplingIt can be shortened by citing Huang et al., 2019, Animal, 9:..Response 1: As suggested by the reviewer, we added the sentence “A detailed description of the location, experimental design and sampling is in the companion paper on the nutritive value of common vetch seed, and are only summarized here” within lines 82-83 of the revised manuscript.
Point 2:L111 Ørskov [16]. Rumen fluids were obtained before morning feeding from 4 adult fistulated dorper. Delete. It had been described above
Response 2: As suggested by the reviewer, we changed “Rumen fluids were obtained before morning feeding from 4 adult fistulated dorper” to “Rumen fluids were obtained before morning feeding” within lines 120-121 of the revised manuscript.
Point 3: L117-118: 6, 8, 12, 24, 36, 48 and 72 h of incubation and corrected for the blanks. The whole process was repeated on a different day. All operations involving rumen fluid were conducted under a…. Only two repetitions. That is an important gap. Most of lab studies are done with three repetitions due to the important factors determining the variability of results (animals, ruminal fluid, moment of assay…). Is it advisable to do always 3 repetitions (3 runs in three different days)
Response 3: For high accuracy, we support reviewers' views. In the future research, we will perform three repetitions according to the reviewer. However, because the test animals have been transferred to other test sites, we are not able to perform supplementary tests now. We are very sorry. López et al. (2005) reported that they used two repetitions to in vitro gas production of feedstuffs (López et al., 2005; Assessment of nutritive value of cereal and legume straws based on chemical composition and in vitro digestibility)
Point 4. L147 Soest et al. [23] using heat‐stable α‐amylase and sodium sulfite. Acid detergent insoluble protein. heat‐stable α‐amylase is not necessary because straw does not have starch
Response 4: From a professional point of view, we considered that the reviewer’s point of view is correct. Before nutrient analysis, we don't know the nutrient content of common vetch straw. Larbi et al. (2011) reported that they added sodium sulphite and α-amylase to determine NDF content of common vetch straw (Larbi et al., 2011 Intra-species variations in yield and quality determinants in Vicia species: 3. Common vetch (Vicia sativa ssp. sativa L.)). Thus, we applied this experimental method. In the future research, we will determine NDF content of common vetch straw according to the reviewer.
Point 5: L173-173: I’m not statistical expert, but the fermentation kinetics parameters should be done with mixed models, considering the variety as fixed effect and the run as random effect.
Response 5: we considered that the reviewer’s point of view is correct. Thus, as suggested by the reviewer, we changed “and means were separated using the Duncan significant difference test at p < 0.05.” to “The gas production parameters were subjected to separate analyses of variance, with varieties as fixed effects and the run as random effect. Variances among means were compared using the Duncan test at p < 0.05” within lines 184-187 of the revised manuscript.
Point 6: Results that are expressed in the Table must not be written in the main text, Please delete all.
Response 6: As suggested by the reviewer, we deleted “The varietal difference between the maximum and minimum values were 379.8 and 2402 kg DM/ha in seed yield and straw yield, respectively” and “The varietal range in harvest index and PUI were, respectively, 13.1 units and 9.8 units”.
Point 7: L231: Lanjian No.2 (22.9 % CHO), and lowest for Lanjian No.3 (20.1 % CHO) and 333A (17.7% CHO).
333A: is it the local variety??? It has not been described before
Response 15: Yes, 333A is the local variety. As suggested by the reviewer, we added information of variety 33A within line 91 of the revised manuscript.
Point 8: 301-305: bean (Vicia faba L.) [29] and chickpea (Cicer arietinum) [31]. The seed yield range (875.2–1255 kg DM/ha) in this study is within ranges reported value of 287–1783 kg DM/ha [8], but lower than ranges reported value of 1340–2240 kg DM/ha [11]. The straw yield range (3154‐5556 kg DM/ha) is higher than reported ranges (629–2226) by Larbi et al. [8], but slightly lower than the range (4620 –305 7320 kg DM/ha) reported by Albayrak et al. [11]. These
It can be sumarized saying that: yields vary between studies (8, 11...) as consequence of the differences in the varieties, agronomic practices and growing conditions (e.g., soil type and climate)....
Response 8: As suggested by the reviewer, we changed “These differences in seed and straw yields between different studies may be partly due to differences in the varieties, agronomic practices and growing conditions (e.g., soil type and climate)” to “ The yields vary between studies (8, 11...) as consequence of the differences in the varieties, agronomic practices and growing conditions (e.g., soil type and climate)” within lines 315-317 of the revised manuscript.
Point 9: L319: 60% DM CP What does it mean??
Response 9: We are very sorry that this is a writing error. As suggested by the reviewer, we changed “threshold content of 60% DM CP”to “threshold content of 6.0% DM CP” within line 328 of the revised manuscript.
Point 10: L323-327: Results of this study showed that ADF, ADL, hemicellulose and cellulose contents of the straws varied from 27.4 to 33.2 % DM, 6.08 to 9.56 % DM, 17.6 to 21.4 % DM, and 21.4 to 23.7 % DM, respectively. Fortina [10] reported higher NDF, ADF, ADL and cellulose (62.8, 46.7, 9.73 and 37.0 %DM) and similar CP and EE contents (10.1 and 0.63 %DM) to our results.
Can be deleted
Response 10: We have made a revision according to referee’s comments.
Point 11: L361-363: The average values for potential gas production observed in the current study are lower than the values reported by López et al. [45] despite the gas 363 production rate was lower than in the this study.
Delete.
Response 11: We have made a revision according to referee’s comments.
Point 12: L363: The differences
Response 12: As suggested by the reviewer, we changed “These differences” to “The differences” within line 367 of the revised manuscript.
Point 13: L372-376: Mean ME value in common vetch straw (6.6 MJ/kg DM) recorded by Abreu and Bruno‐Soares [36] were slightly lower than values in the current study, but similar to values observed by Abreu and Bruno‐Soares [36] and López et al. [45]. Such differences in IVOMD and ME content of common vetch straw among studies may be related with natural variation in common vetch straws themselves, as well as plant growing conditions [8,12,43].
No sense. Rewrite the sentence. It is inconsistent with some sentences wrote before.
Response 13: We have made a revision according to referee’s comments. Please see Line 374-383.
Point 14: The Tables: Add the column of the p values of effect.
Response 14: We have made a revision according to referee’s comments. Please see all Tables.
Additional minor revisions were made to the manuscript using track changes to improve English and readability.
Special thanks to you for your good comments. We hope that the correction will meet with approval.
Round 2
Reviewer 2 Report
Point 3:Two runs are too few for in vitro estudies. I think it is almost mandatory to have 3 runs.
En each run there is a lot of factors affecting results: day, animal, intake, lab, technician... so to reduce all these effects it is mandatory 3 runs.
Point 5: Statistical analysis for fermentation kinetics parameters (A, c and h) should be estimated through a non-linear regression model using NLIN programme. There after as mixed model, as authors did in this second time.
The rest of comments authors had answer correctly